# Breath Analysis for Early Detection of Rising Ketone Bodies in Postpartum Dairy Cows Classified as at Risk of Ketosis

Elaine van Erp-van der Kooij [1,*], Janiek Derix [2], Stijn van Gorp [1], Amy Timmermans [1], Charles Krijnen [1], István Fodor [3] and Liesbeth Dingboom [2]

[1]  Department of Animal Husbandry, HAS Green Academy, Onderwijsboulevard 221, 5223 DE 's-Hertogenbosch, The Netherlands
[2]  Department of Applied Biology, HAS Green Academy, Onderwijsboulevard 221, 5223 DE 's-Hertogenbosch, The Netherlands
[3]   Wageningen Livestock Research, Wageningen University and Research, 6708 WD Wageningen, The Netherlands
[*]  Correspondence: l.verp@has.nl; Tel.: +31-888903600

**Abstract:** Ketosis is a metabolic disorder associated with high production and low energy intake in dairy cows. Fat reserves are mobilized, and gluconeogenesis occurs. Traditionally, rapid tests for blood, milk or urine are used to detect increased ketone body levels in case of ketosis. Breath analysis is relatively new and relevant for the development of automatic sensor systems for early warning. This study aims to determine whether and when a postpartum rise in ketone bodies occurs in breath with elevated blood β-hydroxybutyrate (BHB) in cows at risk of ketosis. Postpartum breath, blood, urine and milk samples were taken daily until day 7, with one prepartum sample excluding milk, and ketone body concentrations were determined. Concentrations of blood BHB were 0.4–2.6 mmol/L (lab) and 0.3–3.1 mmol/L (rapid test), breath acetone was 2.3–20.0 ppm, urine acetoacetate 0–8 mmol/L and milk BHB 0–500 μmol/L. A rise in blood BHB was related to that in urine acetoacetate and milk BHB and followed by a rise in breath acetone. However, breath acetone levels of ketotic and non-ketotic cows were similar. We conclude that rising ketone bodies can be detected in blood, urine, milk and breath, but to use breath analysis as an alternative to rapid tests, longitudinal sampling is needed.

**Keywords:** ketosis; breath analysis; acetone; BHB; acetoacetate; blood; urine; milk

## 1. Introduction

In the past thirty years, milk production of dairy cows has doubled in most countries. The direction of nutrients to the udder immediately after calving has been the basis for successful breeding towards higher milk yield, but there is a large variation in the adaptive responses of individual cows toward energy and nutrient shortages. High milk yields pose a metabolic challenge for the cows, which may result in a decrease in the immune response, reproductive performance and milk quality, as well as cow welfare [1]. Cows with higher milk production are more likely to suffer from ketosis [2], or hyperketonaemia, a metabolic state characterised by elevated concentrations of ketone bodies in blood that diffuse to different body fluids: β-hydroxybutyrate (BHB) can be found in blood and milk, acetoacetate in urine, and acetone in breath [3]. Ketosis occurs mainly postpartum, at the start of lactation, due to a negative energy balance. In that period, the feed consumption, and therefore the energy intake, of the cows do not meet the demands of the high milk production [4]. To compensate for this shortage, the cow breaks down her own body fat, forming ketone bodies. The body can process small quantities of these ketone bodies, but in the case of glucose deficiency, oxaloacetate is being extracted from the tricarboxylic acid (TCA) which leads to an excess of acetyl-Co, causing the formation of more ketone bodies, leading to ketosis. First and second parity cows are less prone to ketosis because they are

less at risk of a negative energy balance [3]. These cows produce less milk and their liver is less fatty [5].

Under normal conditions, ketone bodies are produced by the rumen epithelium from fatty acids in the diet, especially butyrate [6]. Ketone bodies are also produced in the liver from fatty acids, which are mobilised in adipose tissue during negative energy balance [7]. Ketone bodies are important sources of energy [8], especially during negative energy balance and low blood glucose concentrations [7]. At the start of lactation, blood glucose concentration decreases as well as the insulin/glucagon ratio [7], while the concentrations of BHB and non-esterified fatty acids (NEFAs) in blood increase [1,9]. The NEFAs are oxidised by β-oxidation, forming acetyl-CoA, which is used in the TCA cycle [7]. The excess of NEFAs results in an excess of acetyl-CoA that the TCA cycle cannot process, leading to ketogenesis. Acetoacetate is formed from acetyl-CoA, and BHB and acetone from acetoacetate [7]. BHB is the major circulating ketone body in dairy cattle [10]. Ketone bodies are released into the blood where they serve as an energy source for organs such as muscles, brain, mammary glands [6,11], and heart [12]. In short, ketogenesis is a normal physiological process during fat mobilisation, and ketone bodies are important energy sources, but an excess of ketone bodies due to ketogenesis can cause problems in the metabolic process [7]. Risk factors for ketosis are a body condition score > 3.75 (on a scale from 1 to 5), multiple pregnancy, dry period > 70 days, locomotion problems or claw health issues [13].

Due to the negative energy balance during ketosis, the amount of energy available for other processes in the body is decreased [14], and cows have an increased risk of uterine infection, mastitis, fatigue and reduced fertility [15]. Due to the reduced feed intake, there is a higher risk of abomasum displacement [16]. Veterinary treatments, decreased milk production, a prolonged calving interval, and culling can result in high losses for ketotic cows, estimated at approximately 709 or 735 euros per case, and ranging from 64 to 1196 euro [17,18]. In Europe, 16–23% of cows develop ketosis 2–15 days postpartum, with blood concentrations of BHB $\geq$ 1.2 mmol/L [19]. In the Netherlands, 11.2 percent of dairy cows suffer from ketosis in the first months of lactation while the prevalence per farm varies from 0–80% [3].

To detect ketosis at an early stage, ketone bodies can be measured. Several tests are used in practice, to determine BHB in blood, BHB in milk or acetoacetate in urine. Measuring acetone in breath is not yet common practice, partly because no sensors or tests are available yet for routine breath analysis in cows [20].

The measurement of BHB in blood gives the most reliable results in the detection of ketone bodies, because BHB is more stable than acetone or acetoacetate. BHB in blood is therefore considered the gold standard for ketosis diagnosis [21]. Therefore, in this study, we took blood samples, preferably from the jugular vein, the second choice being the coccygeal vein. Acetoacetate, acetone and BHB are small molecules that do not bind to plasma proteins in the blood and therefore can end up in the preliminary urine. When the concentration of ketone bodies in the blood is low, almost all ketone bodies are resorbed in the kidneys [22]. With high concentrations of ketone bodies in the blood, they are filtered out in the Bowman's capsule, resulting in approximately 20% of the ketone bodies being excreted through the urine [23]. Urine test strips measure acetoacetate in the urine, which is a reliable ketosis test and more reliable than milk tests [24,25].

Milk contains acetone, acetoacetate and BHB. The concentration of ketone bodies in milk is about 50% of the total concentration of ketone bodies in the blood [26]. The ketone bodies enter the milk via the blood capillaries surrounding the alveoli, which contain milk forming cells. Ketone bodies can pass through the milk-forming cells into the milk sacks, and are transported with the milk to the milk atrium [23,27]. Milk test strips measure BHB in milk, which has a strong correlation of 0.705 with BHB in blood, and therefore they are suitable for determining ketosis [28]. Blood flows to the capillaries of the lungs, where it releases acetone into the alveoli, after which the air is exhaled. In an earlier study, it was shown that ketosis can be detected by analysing exhaled air from cows [29]. The

acetone concentrations in blood correlated with concentrations of blood BHB (r = 0.81) and milk acetoacetate and acetone (r = 0.70) [29]. In another study, it was shown that acetone concentrations varied between 0 and 14 ppm. Ketotic cows showed higher acetone concentrations in breath, and it was concluded that breath analysis can be a non-invasive way for determining ketosis [30]. Previous research indicated that the rise in acetone levels in breath occur earlier than the rise in BHB in milk and acetoacetate in urine [31]. This was also shown in another study, where acetone was the first rising ketone body in the blood [32].

If acetone is indeed the first ketone body to rise in blood, then acetone in breath might be a good indicator for early detection of rising ketone bodies and risk of ketosis. Through early detection, the cow can be treated in time to prevent a more serious course of disease [33]. However, breath analysis has not yet been compared to blood values. Testing ketosis in blood gives more reliable results than the ketosis test strips for milk and urine [24].

The purpose of this study was to measure ketone bodies in the breath of cows at risk of ketosis just before and after calving, and to compare the rise in ketone bodies in blood, urine, milk and breath. The main research question is whether a rise in ketone bodies is shown in all body fluids and which measurement shows the rise first. If the concentration of acetone in breath increases at the same time or soon after the rise of BHB in blood, then breath analysis is a good alternative for the present ketosis tests. It is non-invasive and there are possibilities to automate the process. This can help the farmer with the early detection and treatment of ketosis, preventing a more serious course of disease, and being beneficial for cow welfare, as well as for the technical and financial results of the farm.

## 2. Materials and Methods

### 2.1. Experimental Design and Animal Ethics

Seven cows from a commercial free stall herd of 117 dairy cows were selected for the experiment before calving. A priori power analysis using GPower (GPower Version 3.1.9.4., University of Kiel, Kiel, Germany) showed that four and eight cows are required for a power of 80 and 95%, respectively. All cows were Holstein Friesian dairy cows, from the same herd and production group and with the same feeding regime. Cows were included in the study when they met at least one of the inclusion criteria: BCS > 3.75 (on a scale from 1 to 5), length of dry period > 70 days, multiple pregnancy, parity > 2, deviating claw health score or deviating locomotion score. General information on lactation number and BCS are shown in Table 1. Cows were not separated from the herd except for calving and if necessary for treatment when ill. Of each cow, one sample of blood, breath and urine was taken one week before the expected day of calving and daily samples of blood, breath, urine and milk were taken during one week after calving. Cows were fed daily at approximately 7:30 a.m. and sampling of milk, urine, breath and blood was done between 8 and 9.30 a.m. To monitor health and welfare of the cows during the study, a health log was filled once a week that described general health of the cow, and a welfare log was used at each sampling that described the sampling process and any events during that particular sampling. The study was performed in accordance with European Community guidelines for the use of experimental animals. Approval of the local committee for animal studies was obtained (AVD7310020198244).

### 2.2. Sampling and Analysis

Sampling of blood, breath, urine and milk was done by two researchers. Before sampling, each cow was tied to the feeding fence, looking to the right, with halter and rope (Figure 1). Sampling was always done in the order milk, urine, breath and blood, because the blood sampling procedure may induce stress which could effect values for ketone bodies in breath, milk or urine. Besides this, it was possible this way to immediately release a cow that did not respond well to the blood sampling procedure, without losing the other samples.

**Table 1.** Cow number, lactation number, dry days and body condition score (BCS, scale 1–5) at calving.

| Cow Number | Lactation | Dry Days | BCS at Calving |
|:---:|:---:|:---:|:---:|
| 12 | 4 | 57 | 4 |
| 13 | 3 | 74 | 3 |
| 29 | 3 | 59 | 3.5 |
| 51 | 2 | 59 | 4.5 |
| 81 | 3 | 78 | 4 |
| 113 | 2 | 80 | 3.5 |
| 337 | 3 | 98 | 4.5 |

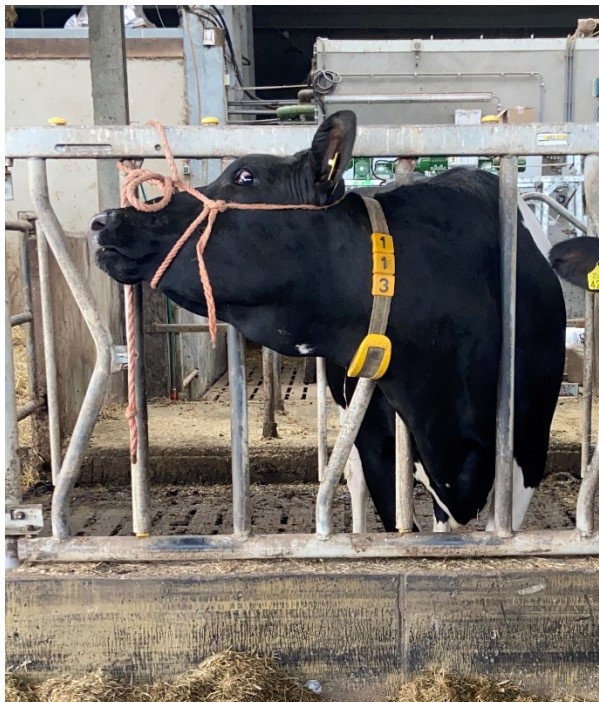

**Figure 1.** cow tied to the feeding fence for sampling.

Two samples of environmental air in the barn were taken at the feeding fence, to determine whether there could be an influence of outside air on the acetone concentration in exhaled air.

2.2.1. Blood Sampling and Analysis

Blood was taken from the jugular vein if possible, and when this did not succeed, from the coccygeal vein, using a standard protocol. A vacuum test tube was filled, and these samples were sent to the Royal GD (Deventer, the Netherlands) to be tested for BHB.

After filling the vacuum test tube with blood for the lab, a drop of blood was used for the rapid test. The Precision Xtra meter (Abbott Diabetes Care, Saint-Laurent, QC, Canada) and the corresponding test strips were used. The sensitivity of this test is 88 and 96%, and its specificity is 96 and 97%, at 1.200 and 1.400 mmol BHB/L whole blood, respectively [24]. A drop of blood was applied to the test device, which allowed the reading of the BHB concentration in the blood after 10 s. A threshold of $\geq 1.4$ mmol/L BHB in blood (based on laboratory results) according to Suthar [15] was used to classify a cow as ketotic. We also took into account the opinion of the veterinarian, who decided whether to treat a cow for ketosis based on the test results.

2.2.2. Breath Sampling and Analysis

Breath samples were taken using a nostril sampler with an FTFE tube connected to a nalophan bag (10 L) which stored the exhaled air (Figure 2). The sampler has a diameter of 4.5 cm, a length of 8.5 cm and is fitted with a non-return valve. The hole at the bottom of the sampler is attached to a nalophan bag (MediSense, Groningen, the Netherlands) [34]. Samples were collected by fitting the nostril sampler to one nostril and closing the other nostril while the cow exhales (Figure 3). Ructuses were palpated manually by the researchers and were not included in the breath sample. After sampling, the bag was sealed from environmental air. When opening the bag to connect the Kitagawa test tube, a little breath sample was forced out of the bag to prevent contamination of the collected breath sample by the outside air.

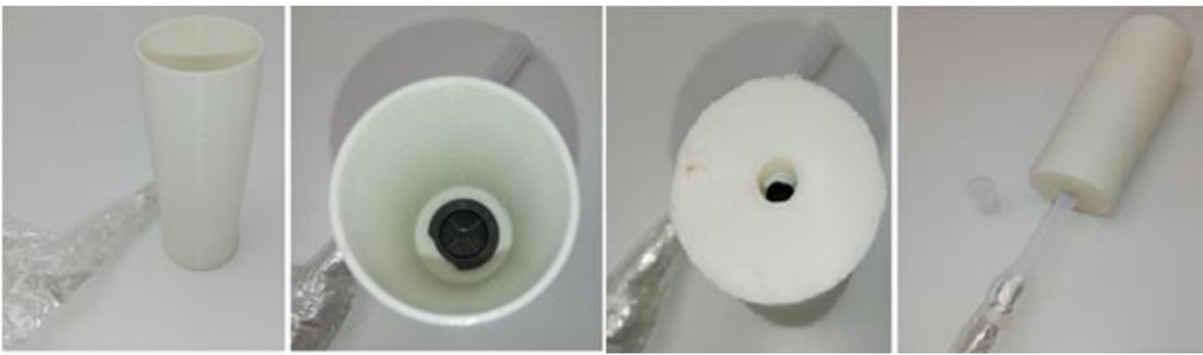

**Figure 2.** Nostril sampler connected to Nalophan bag.

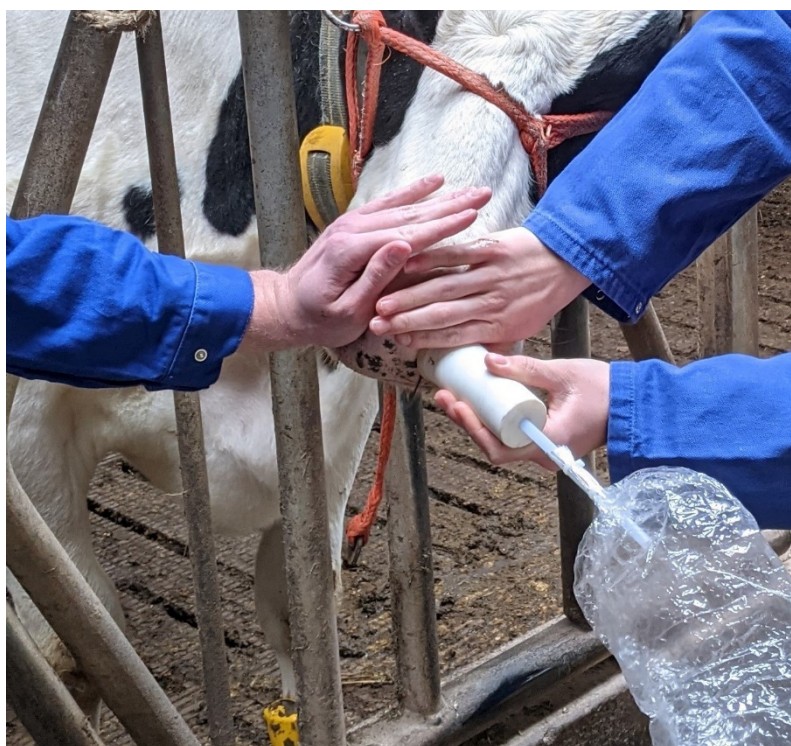

**Figure 3.** Collection of a breath sample with the nostril sampler and the nalophan bag attached; while taking breath samples, one researcher checked for ructuses at the side of the neck (not shown in the picture).

The acetone concentration in the exhaled air was measured with the Kitagawa gas detector fitted with acetone 102SD tubes with a measurement range of 20–5000 ppm per pump stroke of 100 mL. Two gas detection tubes were used per bag and ten pump strokes

per tube were performed. Acetone concentrations were read after each pump stroke. Since it was unknown whether the acetone in the nalophan bag would be homogeneously distributed, the bag was gently shaken before the acetone concentration was determined by the Kitagawa test. The acetone concentration was calculated as the average from two test tubes, estimated using a polynomial trend line in MS Excel Version 2204 (Redmond, WA, USA). This was done in a stepwise procedure for each test tube: (1) acetone concentration was measured by performing ten pump strokes with the Kitagawa test device. This gave ten read values for pump strokes 1–10. (2) These read values were put in a graph and a polynomial trend line was drawn. The formula of the trendline was calculated using Excel. (3) According to the manufacturer, pump stroke 2 should be used. The estimated reading on pump stroke 2 was calculated by putting an x-value of '2' in the trendline formula. (4) The ppm concentration for this estimated read value is calculated by multiplying the estimated read value at pump stroke 2 with $0.4^{(\text{pump strokes}-1)}$, as described by the manufacturer. An example of this calculation can be found in Figure 4.

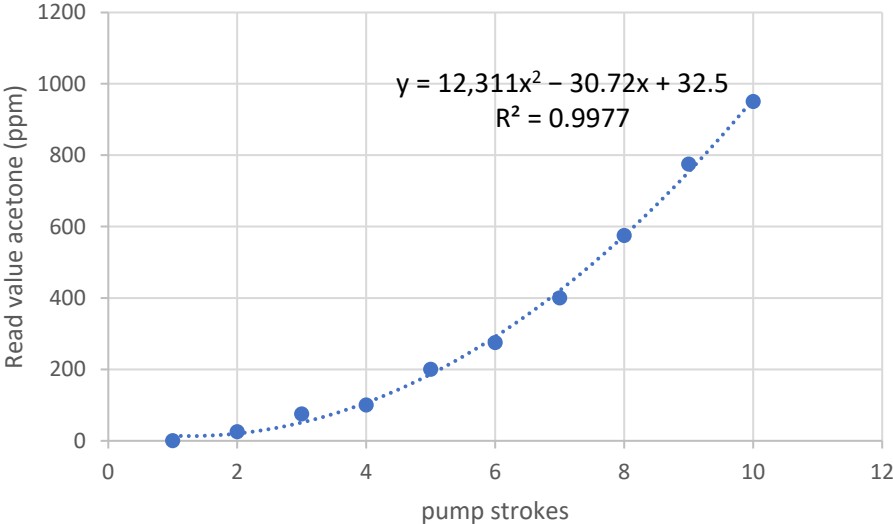

**Figure 4.** Example of the calculation of ppm values for acetone in breath from read values. The expected read value at pump stroke 2 in this example is $12.311 \times (2)^2 - 30.72 \times (2) + 32.5 = 20.304$; calculated acetone concentration for this sample is $20.304 \times 0.4^{(2-1)} = 8.12$ ppm.

In the field study, all acetone measurements were duplicates, using one bag per cow and two tubes per bag. Validation of the acetone measuring tests was done using triplicates: three bags of exhaled air were tested with three test tubes each.

### 2.2.3. Urine Sampling and Analysis

Urine samples were taken by massaging the perineum of the cow to stimulate urinating behaviour. Urine was collected in a cup and tested with a test strip at the farm. The concentration of acetoacetate in urine was measured by using Ketostix test strips (Ascensia Diabetes Care, Mijdrecht, The Netherlands). These strips have a reported sensitivity of 78% and specificity of 99% [35]. Acetoacetate concentration in the urine was indicated by the colour of the test strip, and fell into either one of the following categories: 0.0, 0.5, 1.5, 4, 8 and 16 mmol/L.

### 2.2.4. Milk Sampling and Analysis

Milk samples were collected by careful manual milking, using a squeezing motion. After discarding the first few milk jets, the milk sample was collected in a cup and tested with a test strip at the farm. The concentration of BHB in milk was determined by using a PortaBHB milk ketone test strip (Elanco Animal Health Nederland, Utrecht, The Netherlands). Colour of the test strips indicated a level of milk BHB in one of the following categories: 0, 50, 100, 200, 500 and 1000 μmol/L.

### 2.3. Statistical Analysis

All analyses were performed in IBM SPSS Statistics version 26 (IBM, Armonk, NY, USA) [36]. We used a repeated measures linear model in General Estimating Equations, with cow number as subject and days postpartum as within-subject variable. This allowed for the individual cow to be used as the unit of analysis and accounted for the repeated sampling. The dependent variables were either acetone in breath, BHB in milk, or acetoacetate in urine, and the predictor was BHB concentration in blood, analysed in the laboratory. The repeatability of the test results in the validation test was determined by the intraclass correlation coefficient (ICC) [37]. A Pearson correlation test as well as a repeated measures linear model in General Estimating Equations, with cow number as subject and days postpartum as within-subject variable, were used to test the agreement of the blood BHB rapid test and laboratory results, with laboratory results as the dependent variable and rapid test results as the predictor. An independent *t*-test was used to compare postpartum acetone levels of ketotic and non-ketotic cows. The level of significance was set to $p < 0.05$.

## 3. Results

### 3.1. General Results

The seven selected cows in the study calved between 3 and 17 days after the prepartum sampling, which was scheduled approximately one week before the expected calving date. Calving dates ranged from −4 days before to +14 days after the expected calving date. Cows were in their 3rd, 4th or 5th parity, and the length of the dry period ranged from 57 to 98 days. No cows were lame or showed any clinical signs before calving. No health issues resulted from the sampling procedures.

The concentration of BHB in blood ranged from 0.4 to 2.6 mmol/L for the lab tests and from 0.3 to 3.1 mmol/L for the rapid test. The concentration of acetone in breath ranged from 2.3 to 20.0 ppm, acetoacetate in urine from 0 to 8 mmol/L and that of BHB in milk from 0 to 500 µmol/L. For one animal, we did not manage to get enough blood sample for the laboratory analysis, due to the wild character of the cow. Although, we could draw enough blood sample to perform the rapid tests. In Table 2, results are shown for all cows of the blood lab test. For this cow we estimated the blood lab results from the rapid test results, with the coefficients of the model that was used to test the agreement of the rapid and lab tests (Figure 5).

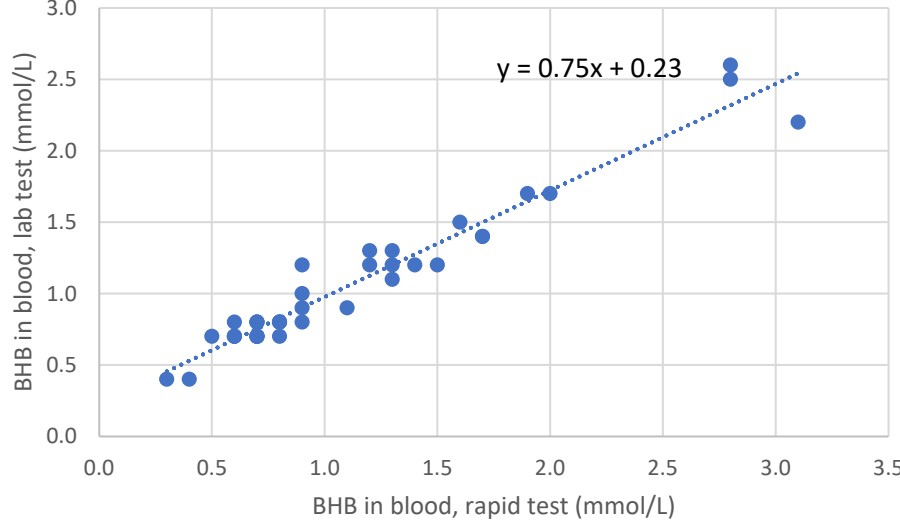

**Figure 5.** Relationship between blood BHB analysed with a rapid test or in the laboratory ($p < 0.001$).

Four of the seven cows developed ketosis according to the criteria of BHB in blood (threshold 1.4 mmol/L). These cows were also ketotic according to at least one urine test (urine acetoacetate ≥ 1.5 mmol/L) and were treated by the veterinarian. Two of the four cows that were classified as ketotic also had positive test results from the milk test (BHB in

milk $\geq$ 200 µmol/L). Of the three cows that were classified as non-ketotic according to the blood BHB level, one showed a urine test that indicated ketosis (1.5 mmol/L acetoacetate) but all milk tests showed lower milk BHB (Table 2). We decided to classify the cows as ketotic using the blood test threshold only, since this corresponded with the veterinarian's decision to treat the cows.

**Table 2.** Descriptive statistics of ketone bodies in blood, breath, urine and milk samples, and the ketosis status for all cows in the study ($n$ = 7).

| Cow Number | Ketone Body and Sample Type | Mean $\pm$ SD | Min, Max., Samples | Classified as Ketotic (K) or Non-Ketotic (Non-K) |
|---|---|---|---|---|
| 12 | BHB in blood | 1.1 $\pm$ 0.4 | 0.4, 1.7, 6 | K |
| | Acetone in breath | 8.3 $\pm$ 4.6 | 2.3, 16.2, 7 | |
| | Acetoacetate in urine | 1.1 $\pm$ 1.4 | 0.0, 4.0, 7 | |
| | BHB in milk | 64 $\pm$ 47 | 0, 100, 7 | |
| 13 | BHB in blood | 0.9 $\pm$ 0.4 | 0.4, 1.5, 6 | K |
| | Acetone in breath | 8.0 $\pm$ 2.6 | 4.3, 11.0, 8 | |
| | Acetoacetate in urine | 0.8 $\pm$ 0.7 | 0.0, 1.5, 9 | |
| | BHB in milk | 69 $\pm$ 37 | 0, 100, 8 | |
| 29 | BHB in blood * | 0.9 $\pm$ 0.2 | 0.5, 1.1, 8 | Non-K |
| | Acetone in breath | 12.5 $\pm$ 3.6 | 7.3, 17.2, 9 | |
| | Acetoacetate in urine | 0.3 $\pm$ 0.3 | 0.0, 0.5, 8 | |
| | BHB in milk | 43 $\pm$ 53 | 0, 100, 7 | |
| 51 | BHB in blood | 0.7 $\pm$ 0.05 | 0.7, 0.8, 6 | Non-K |
| | Acetone in breath | 7.1 $\pm$ 2.1 | 5.3, 10.9, 6 | |
| | Acetoacetate in urine | 0.1 $\pm$ 0.2 | 0.0, 0.5, 6 | |
| | BHB in milk | 70 $\pm$ 45 | 0, 100, 7 | |
| 81 | BHB in blood | 2.0 $\pm$ 1.0 | 0.8, 2.6, 3 | K |
| | Acetone in breath | 10.7 $\pm$ 2.3 | 7.5, 13.8, 8 | |
| | Acetoacetate in urine | 3.8 $\pm$ 3.7 | 0.0, 8.0, 9 | |
| | BHB in milk | 114 $\pm$ 38 | 100, 200, 7 | |
| 113 | BHB in blood | 1.3 $\pm$ 0.5 | 0.7, 2.2, 9 | K |
| | Acetone in breath | 14.3 $\pm$ 2.0 | 9.8, 16.7, 9 | |
| | Acetoacetate in urine | 2.0 $\pm$ 2.5 | 0.0, 8.0, 9 | |
| | BHB in milk | 138 $\pm$ 151 | 0, 500, 8 | |
| 337 | BHB in blood | 0.9 $\pm$ 0.2 | 0.7, 1.3, 8 | Non-K |
| | Acetone in breath | 10.7 $\pm$ 4.7 | 6.7, 20.0, 7 | |
| | Acetoacetate in urine | 0.5 $\pm$ 0.5 | 0.0, 1.5, 8 | |
| | BHB in milk | 71 $\pm$ 49 | 0, 100, 7 | |

* estimated from rapid test.

The cows that were classified as ketotic were treated for ketosis with energy pills, propylene glycol, and/or a supplement containing energy, calcium and vitamins, according to the farmer and veterinarian's decision. One cow was treated for milk fever with Ca pills and a Ca/Mg infusion. One of the cows developed ketosis despite having received a monensin bolus three weeks before calving as a preventive measure.

### 3.2. Validation of the Acetone Measuring Method

Three bags of breath sampled from the same cow were used three times, to determine acetone levels in a total of nine test tubes. Read values at each pump stroke were used to calculate acetone levels at two pump strokes. The calculated acetone concentration in the validation test ranged from 5.2–11.4 ppm with an average of 7.9 ± 2.1 ppm. The repeatability of test results in the validation test was excellent, with an ICC of 0.994. The acetone level of environmental air, as measured near the feeding fence, was 0.77 ± 0.30 ppm.

### 3.3. Validation of the Rapid Test for Blood BHB

Results from the rapid test and the lab test for blood BHB were highly related, with a Pearson correlation of 0.971 ($p < 0.001$, Figure 5), with the results from the rapid test being slightly higher than from the lab test (B = 0.75). In this study, we used results from the lab tests as the gold standard for BHB in blood.

### 3.4. Relation between Blood BHB and Ketone Bodies in Breath

The average breath acetone concentration from six cows before calving was 10.4 ppm, ranging from 2.3 to 20 ppm. For one cow, no breath sample was taken before calving due to a late delivery of test materials. Due to missing values and the variable time to calving, we did not use these breath sample results in the analysis.

The concentration of acetone in breath samples and the corresponding BHB concentrations in blood are shown in Figure 6. The rise in breath acetone concentration was related to a rise in blood BHB on the same day ($p = 0.047$) and to a rise in blood BHB of the day before ($p = 0.010$). This suggests that a rise in ketone bodies in breath follows a rise in ketone bodies in blood. No relation was found between acetone in breath and blood BHB of the day after ($p = 0.06$). This means that a rise in acetone does not precede a rise in blood BHB.

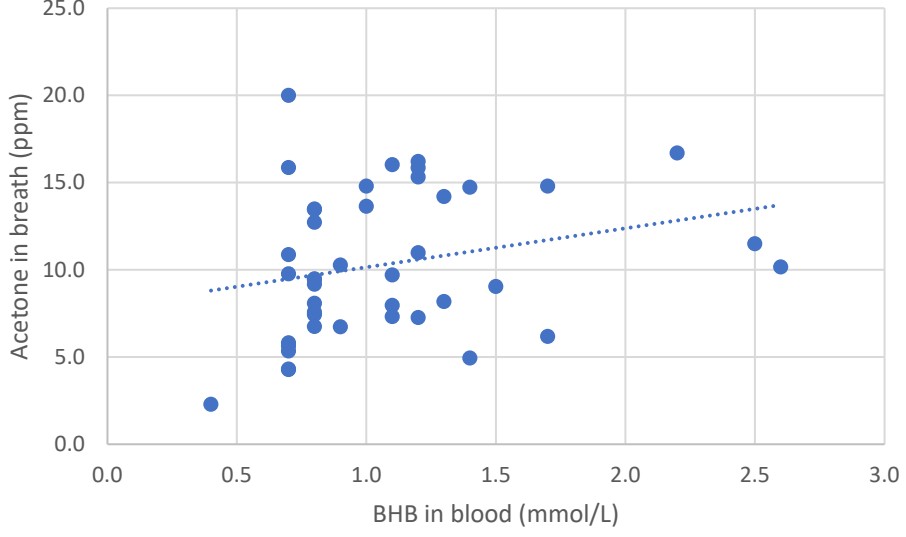

**Figure 6.** scatter plot of blood BHB and acetone in breath for all cows in the study ($n = 7$). Model equation. $y = 2.65x + 7.02$.

When split in two groups, we found a significant relationship between acetone in breath and BHB in blood for ketotic cows ($p = 0.03$, $y = 2.64x + 7.14$) but not for non-ketotic cows ($p = 0.54$, $y = -2.72x + 12.76$). Acetone concentrations in breath and BHB in blood after calving are shown in Figure 7a,b for four ketotic and three non-ketotic cows, respectively.

There was no difference in mean acetone concentrations between ketotic and non-ketotic cows. The mean (±SD) breath acetone concentration postpartum was 10.9 ± 3.6 and 9.9 ± 3.4 for ketotic and non-ketotic cows, respectively ($p = 0.34$).

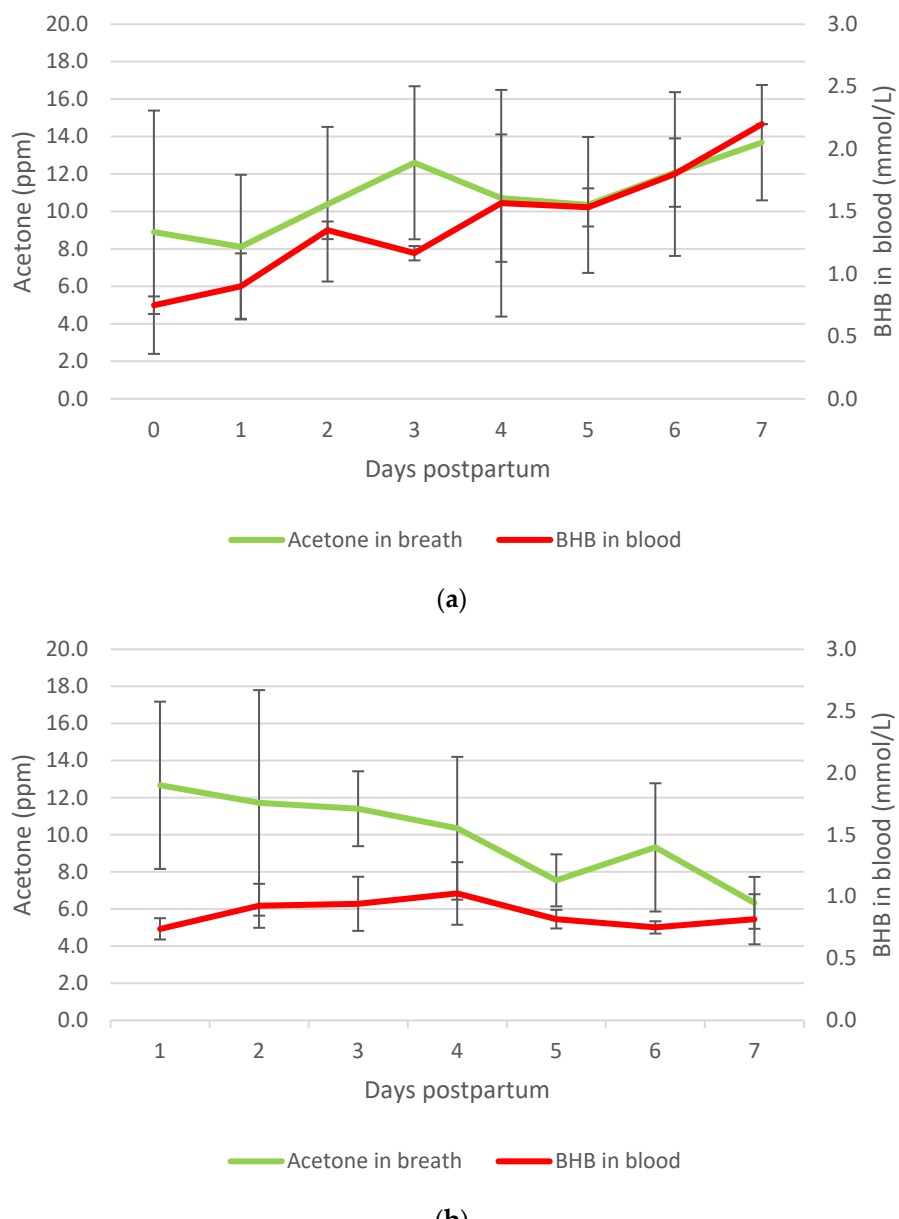

**Figure 7.** (**a**) Acetone concentrations in breath and BHB concentrations in blood (±SD) for four ketotic cows, daily samples until seven days postpartum. (**b**). Acetone concentrations in breath and BHB concentrations in blood (±SD) for three non-ketotic cows, daily samples until seven days postpartum.

*3.5. Relation between Blood BHB and Ketone Bodies in Urine and Milk*

A rise in acetoacetate in urine was related to a rise in BHB in blood on the same day (B = 4.2, $p < 0.001$) as well as to a rise in blood BHB of the day before (B = 3.22, $p < 0.001$) or the day after (B = 1.38, $p < 0.001$). A rise in BHB in milk was related to blood BHB on the same day (B = 98.3, $p = 0.03$), to a rise in blood BHB the day after (B = 21.5, $p = 0.03$), but not to BHB in blood the day before ($p = 0.29$). This means that a rise in urine acetoacetate and a rise in milk BHB occurred at the same time or just after the rise in blood BHB. Results for BHB in blood, BHB in milk and acetoacetate in urine are shown in Figure 8 for ketotic cows and Figure 9 for non-ketotic cows.

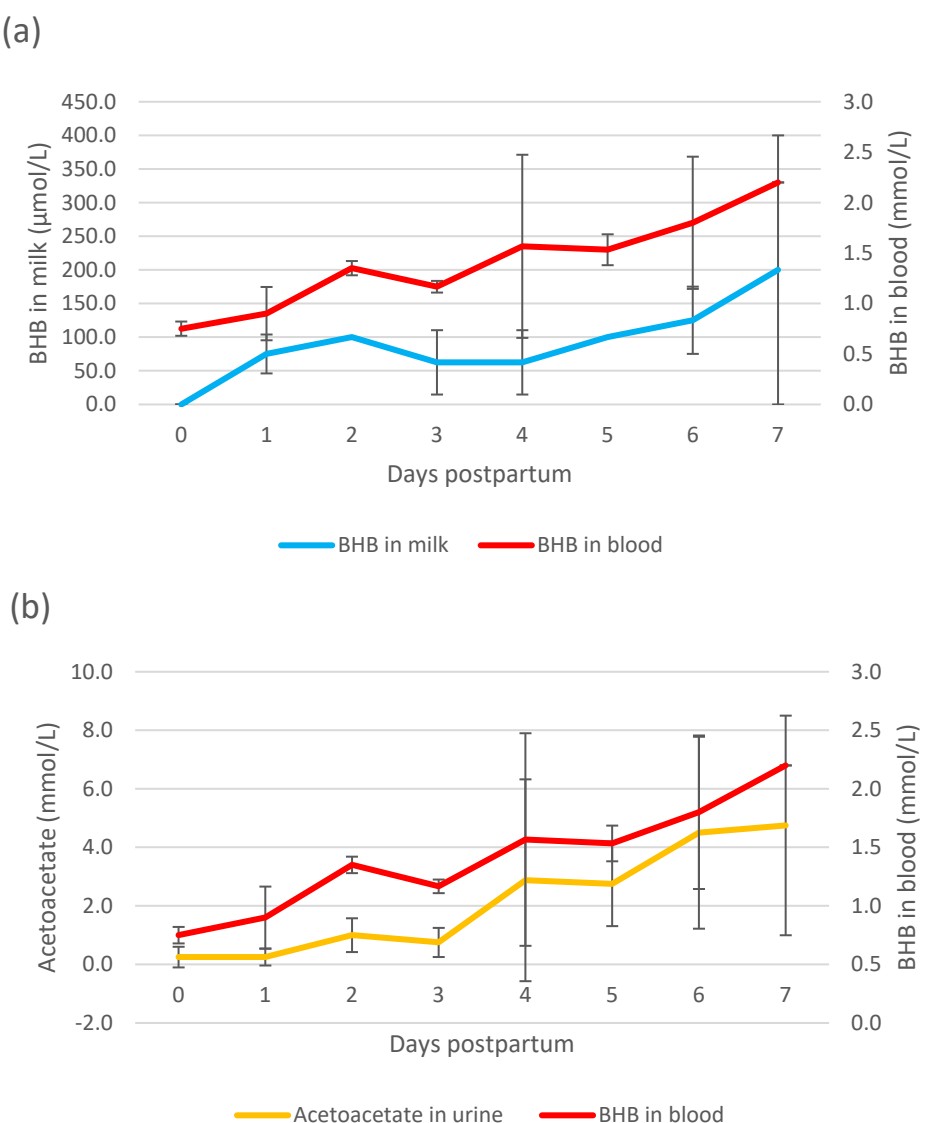

(a)

(b)

**Figure 8.** Concentration of BHB in blood and BHB in milk (**a**) and acetoacetate in urine (**b**) (±SD) for four ketotic cows, daily samples until seven days postpartum.

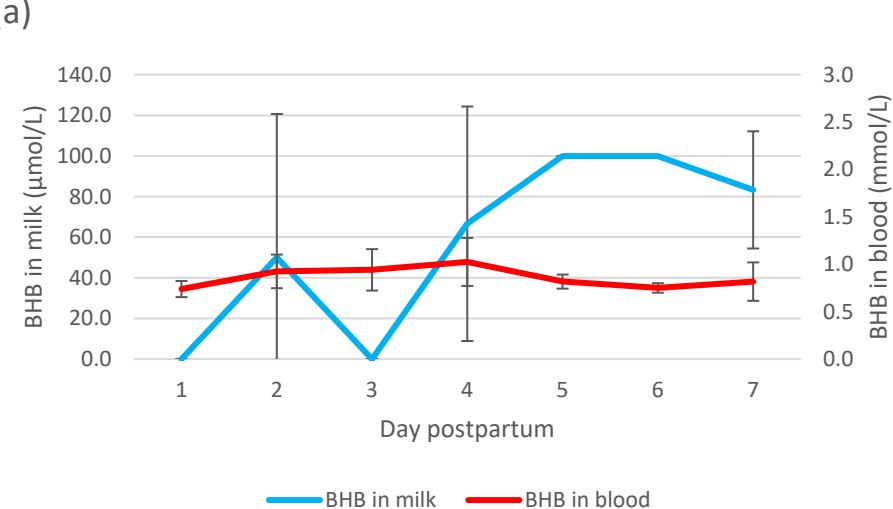

(a)

**Figure 9.** *Cont.*

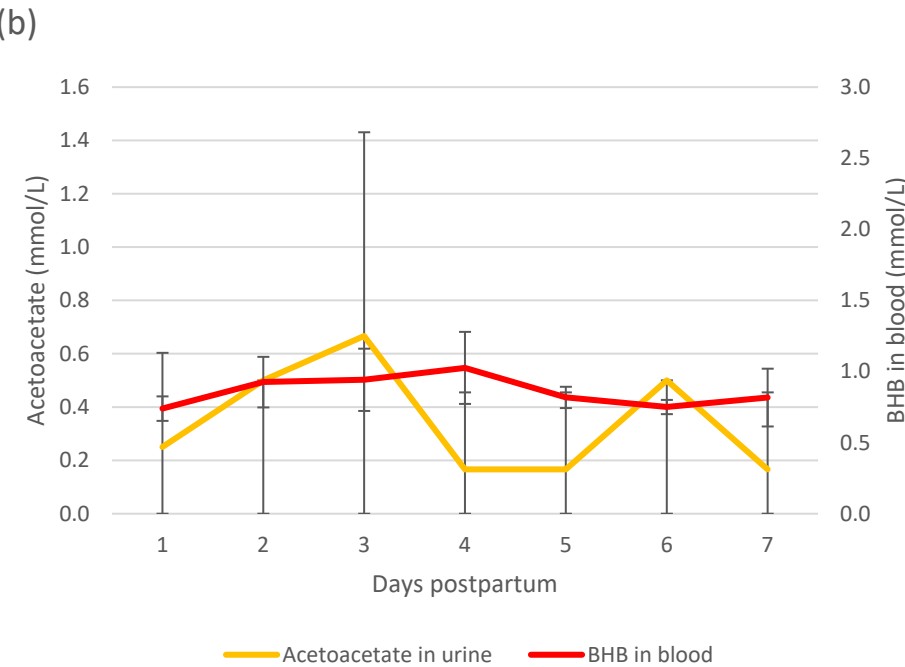

**Figure 9.** Concentration of BHB in blood and BHB in milk (**a**) and acetoacetate in urine (**b**) (±SD) for three non-ketotic cows, daily samples until seven days postpartum.

## 4. Discussion

The aim of this study was to determine the relation between the postpartum rise in ketone bodies in blood, breath, milk and urine of dairy cows at risk of ketosis. During the process of ketosis, when the body derives energy from fat molecules instead of drawing it from glucose, ketone bodies are formed when the liver rapidly metabolizes fatty acids into acetyl-CoA. These ketone bodies, BHB, acetoacetate and acetone, can be detected in blood and are excreted in breath, milk and urine [19]. It is therefore to be expected that blood levels of ketone bodies rise first, and subsequently the levels of ketone bodies in breath, urine and milk rise when a cow enters a negative energy balance. However, since BHB and acetone are formed from acetoacetic acid, it is possible that these three ketone bodies do not rise at the same time and to the same levels in blood. If blood levels of acetone rise sooner than blood levels of BHB, then acetone might rise sooner in breath than BHB rises in blood. In a study by Mottram and others, acetone in the breath increased a few days earlier than BHB in blood [38]. In our study however, we found that acetone in breath follows the rise in BHB in blood in time. Acetone in breath was positively related to blood BHB concentrations of the same day and to blood BHB of the day before. These results are in line with results from studies that found a positive relation between acetone in breath and BHB in blood in human patients [39] and dairy cows [30], but in contrast to another study on dairy cows, where acetone in breath was increased a few days earlier than BHB in blood [38].

Seven cows were followed in this study, which leads to a power of 91%, but is still a limited number. Using more cows might have led to stronger conclusions, since there was a large variation between the results of individual cows. The main purpose of our study was to study the patterns of ketone bodies within cows and not between cows, but due to the variation of these patterns, conclusions were more difficult to draw.

Four out of seven selected cows in this study were classified as ketotic, with blood BHB levels rising above 1.4 mmol/L. Measurement of blood BHB values is the gold standard test for the diagnosis of subclinical ketosis. β-hydroxybutyrate has higher concentrations in blood than acetone. The amount of BHB differs per vein; on average 0.3 to 0.4 mmol/L less BHB is found in the subcutaneous abdominal vein than in the jugular vein and in the coccygeal vein. This might be due to BHB being absorbed in the udder and used by the

cow to produce milk fat [40]. No differences have been found between the concentrations of BHB in the jugular vein and in the coccygeal vein [41]. The most commonly used cut-off points for subclinical ketosis are ≥1.2 mmol/L and ≥1.4 mmol/L [21,41]. For humans, cut-off points are mentioned for blood BHB of ≥0.3 mmol (slightly elevated ketones), ≥0.5 mmol (onset of nutritional ketosis), and ≥1.0–1.5 (higher level of nutritional ketosis), usually a threshold is not used, but the patterns are evaluated and used for diagnostic reasons [39]. For cows, veterinarians use a threshold value but combine it with the manual inspection and assessment of the cow's health, before taking a decision on treatment. In this study, the professional opinion of the farm's veterinarian supported our decision to use a threshold value of 1.4 mmol/L: the veterinarian did not consider the cows that showed levels of 1.2 and 1.3 mmol/L ketotic, and did not treat them. We had expected this high number of cows with BHB levels above the threshold, since ketosis is a common disease in high producing cows, a prevalence of 22–60% of subclinical ketosis has been reported for European farms [17,29,42], and because cows were specifically selected based on ketosis risk. The cows classified as ketotic showed a postpartum rise in blood BHB and also a rise in acetone in breath, acetoacetate in urine and BHB in milk. The cows that were classified as non-ketotic did not show rising patterns in ketone bodies.

Although the rise in acetone in breath was related to the rise of BHB in blood, we found that mean levels of acetone for ketotic and non-ketotic cows did not differ. In earlier studies, the acetone concentration in blood increased from $70 \pm 20$ μmol/L ($3.8 \pm 1.1$ ppm) in healthy cows to $420 \pm 70$ μmol/L ($24.2 \pm 3.8$ ppm) in cows with subclinical ketosis [43] and the level of acetone in breath was 0–2 ppm for non-ketotic and to 4–13 ppm for ketotic cows [30,38]. All cows in the present study showed acetone concentrations > 4 ppm. This is higher than expected from the aforementioned studies but can be explained by the fact that we selected cows with a higher risk of ketosis, and because there were differences in production levels, feeding, sampling, and diagnostic tools compared to the older studies. Breath acetone concentrations before calving were also high for some cows. Acetone can occur in rumen fluid [44] and may evaporate, increasing the concentration of acetone in the exhaled air when a cow eructates [45]. We therefore took care to not sample breath when the cow eructated. However, it is possible that in some samples, rumen-derived exhaled air may have been included causing a higher acetone concentration. Because of the large variation and elevated levels of acetone in breath before calving, absolute levels of breath acetone concentration cannot be used as a detection method for ketosis. So, in order to detect ketosis, longitudinal sampling is needed, because only then the rising pattern in breath acetone can be detected for an individual cow. This is in line with results from human studies, where it is advised to take several samples of breath or blood per day, because of the large variation. They state that multiple samples are needed in order to record the daily ketone dynamics [39].

Blood sampling was done approximately one hour after feeding, between 8 and 9:30 am. This had practical reasons, since cows were standing at the feeding rack at that time and could easily be tied up, avoiding unnecessary stress due to catching and separating cows. However, there is a known postprandial rise in BHB in blood a few hours after feeding, due to the increase in butyrate in the rumen after eating fresh feed, and the higher conversion rate of butyrate to BHB in the rumen wall [41,46], and this rise may vary due to varying amounts of feed intake between the different cows. By sampling the cows at the same time, we decreased the differences in postprandial rise due to time. This is in line with sampling methods in other studies, sampling at 9 a.m. [41], or 'in the morning' [38,42] and usually after feeding. It is also known that age or parity can influence BHB levels in blood, with cows in higher parities showing higher blood BHB levels and being more at risk for ketosis [47]. In this study we did not use primiparous cows, since we were looking for cows that would show a rise in ketone bodies. However, due to the small number of cows, we cannot test the effect of parity or age on the blood BHB levels.

BHB concentration in blood was analysed using a rapid test and in the laboratory. Results from the rapid test correlated well with those of the laboratory. This means that it

would be advisable to use this test as an alternative for the blood sample for the lab. Only a drop of blood is needed instead of a full test tube, which is much easier to obtain and less of a welfare issue for the cow. Our experience with the rapid test is similar to those reported in the literature [21]. Concentrations of BHB in milk and acetoacetate in urine followed the values of blood BHB and showed high levels in ketotic cows and low levels in non-ketotic cows, as expected [21].

We analysed acetone in breath with a handmade nostril sampler and a commercial Kitagawa test device. The range of breath acetone concentrations in this study was 2.3–20.0 ppm. This is comparable to the results of earlier studies that found ranges of 0–14 ppm [30], and an average of $8.1 \pm 3.1$ ppm for non-ketotic cows and of $17.6 \pm 4.6$ ppm for ketotic cows [38]. The acetone level in environmental air near the feeding fence was low. Previous research has shown that the concentration of acetone in environmental air can be as high as 7 ppm [38], which might increase breath acetone concentration even in non-ketotic cows. Before calving, we found high acetone concentrations in breath of two cows out of the six that were sampled, but in these cases only single samples were drawn instead of duplicates. There seemed to be no relationship between these results and the number of days to calving: high breath acetone concentrations were found for cows that were sampled on days $-8$ and $-6$ before calving, while low concentrations were found for the other cows that were sampled on days $-21$, $-10$, $-4$ and $-3$ before calving. An explanation could be that ructuses may have gone unnoticed, raising the acetone concentration in the sample [45]. The sampling method of breath acetone was validated and produced reliable results. A calculation method was developed based on the trendline of read values of ten pump strokes, to compensate for misreadings, especially when acetone levels are low and hard to read from the test tube. However, we worked with a hand-made nostril sampler, bags, test tubes and a pump, which made the procedure complicated and labour-intensive. It would save much time and effort if a sensor would be developed for acetone in breath, with a device that automatically sampled exhaled air. Similar solutions have been developed to monitor individual methane emission, using either portable "sniffers" in milking robots [48,49], or standalone devices using feed bait to attract cows and active airflow to sample breath for methane measurement [50].

## 5. Conclusions

This study shows that ketone levels increase not only in urine and milk, but also in breath samples at the same time or shortly after a rise in blood ketone levels in postpartum dairy cows at risk of ketosis. An advantage of breath analysis is the non-invasive sample collection, which offers potential for automated breath sampling and analysis in the future. We suggest therefore that breath analysis can be used for early detection of a rise in ketone bodies as a non-invasive alternative to blood tests. In order to use breath analysis as an alternative to urine or milk tests, longitudinal sampling is needed to find the rising pattern of acetone in breath. Further research should be aimed at the practical application and automation of such a breath-based test. In addition, the criteria to assess the risk of ketosis based on breath acetone patterns should be determined.

**Author Contributions:** Conceptualization, L.D. and E.v.E.-v.d.K., methodology, L.D., E.v.E.-v.d.K., I.F., J.D., S.v.G. and A.T., formal analysis, E.v.E.-v.d.K. and C.K., investigation, L.D., E.v.E.-v.d.K., J.D., S.v.G. and A.T., data curation, J.D., S.v.G. and A.T., writing—original draft preparation, E.v.E.-v.d.K., writing—review and editing, E.v.E.-v.d.K., I.F., C.K., J.D., S.v.G. and A.T. All authors have read and agreed to the published version of the manuscript.

**Funding:** István Fodor was supported by the TKI Agri & Food project LWV19143 and the partners Melkveefonds and Connecterra.

**Institutional Review Board Statement:** The animal study protocol was approved by the Ethics Committee of Utrecht University (AVD7310020198244, approval date 27-1-2022).

**Informed Consent Statement:** Not applicable.

**Data Availability Statement:** Data from this experiment will be made available upon request by contacting Lenny van Erp at l.verp@has.nl.

**Acknowledgments:** We are grateful to Wilfred de Bruijn for the opportunity to work with his cows, and to veterinarian Bert van Niejenhuis of DG Rivierenland for providing us with a practical training to take blood samples and to arrange for the lab tests. We thank Ascensia Diabetes Care for sponsoring the urine test strips and Elanco Animal Health for sponsoring the milk test strips.

**Conflicts of Interest:** The authors declare no conflict of interest.

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
