# Peer review of "Breath Analysis for Early Detection of Rising Ketone Bodies in Postpartum Dairy Cows Classified as at Risk of Ketosis"

_ruminants, doi:10.3390/ruminants3010005_

Round 1

Reviewer 1 Report (Previous Reviewer 2)

The questions are carefully addressed. 

Author Response

Thank you for your time and feedback. 

Reviewer 2 Report (Previous Reviewer 3)

The authors have addressed all the minor suggestions. However, in my opinion the major flaw in the experimental design remains the lack of a control group of cows not at risk of ketosis. As the authors affirmed, ketogenesis is a normal physiological process, in particular in cows with large milk production which can meet a serious NEB condition, such as Holstein Friesian cows. So the lack of a baseline value does not allow to understand the entity of the breath acetone increase with the insurgence of the ketosis (in particular in presence of high SE as shown in the graphs) compared to the BHB blood, which ranges are well known. This reduces the novelty of the study because, as the authors imply at the end of the paper, the criteria to assess the risk of ketosis based on breath acetone patterns have not been determined yet.

However, the results showed that (at least in confirmed ketotic cows) the rise in ketone bodies in breath follows a rise in ketone bodies in blood, respecting the aim of the paper to compare the rise in ketone bodies in blood, urine, milk and breath. The entity of the correlation between breath acetone and blood BHB is not well explained, missing a regression formula or a regression coefficient in  Fig 6. I would suggest also showing the regressions within the 2 groups K and not-K.

For this reason, I suggest reconsidering the present paper after changing the title (in my opinion misleading) and including in the title and aims that the comparison of rises in ketone bodies has been conducted in cows classified at risk of ketosis.

I also suggest checking the order of the references.

Round 2

Reviewer 2 Report (Previous Reviewer 3)

I don't have any other suggestions or comments.

This manuscript is a resubmission of an earlier submission. The following is a list of the peer review reports and author responses from that submission.

Round 1

Reviewer 1 Report

Article very well written. I just would like to point out that, considering milk sampling (Material and Methods), maybe a complete milking of cows could provide better samples for ketosis diagnosis.

Anyway, the article is ready for publication. Congrats!

Reviewer 2 Report

The manuscript “Breath analysis for early detection of rising ketone bodies in postpartum dairy cows used a non-invasive nostril sampler to analyze the rising ketone bodies in postpartum dairy cows. The study aims to determine whether and when a postpartum rise in ketone bodies occurs in breath with elevated blood β-hydroxybutyrate (BHB). The authors are studying an interesting and important question, as a better diagnosis of the risk of ketosis. However, several critical questions needs to be addressed as follows. Further studies by recruiting more postpartum cows were suggested to be investigated, even more prepartum cows.

Major flaws:

1. From the data shown in Table 1, it seemed like there is no corrrelation between BHB in blood (as the criteria) and acetone in breath, for example, cow No. 12 and cow No. 29, and also cow No. 81 and No. 337, got the same value of acetone in breath (16ppm and 14ppm), but they were classified into K and non-K groups. Although the authors suggested that longitudinal sampling is needed, the discrepancy above probably could not be solved by multiple sampling.

2.  As described “ Cows were included in the study when they met at least one of the inclusion criteria: BCS > 3.75 (on a scale from 1 to 5), length of dry period > 70 days, multiple pregnancy, parity >2, deviating claw health score or deviating locomotion score, the enrolled 7 postpartum cows in the study have different genetic and/or feeding background, which might make the breath analysis more complicated. Because cows aged has higher parity number might have different value of postpartum BHB with cows experienced fewer parity number, although at the same postpartum day. Whats more, the milk/urine/breath and blood samplings were done after feed in the morning, whether it would be better if samplings were done before feed?  

3.  It is unreasonable to artificially separated and showed in different figures (Figure 6 and Figure 7).

Reviewer 3 Report

The present paper reports an interesting approach to a new to detect the rise of ketosis in the herd using the breath test for the acetone correlated to the BHB levels in the blood. The topic is interesting and not well explored by previous literature. So, the purpose and significance of this research are clear. However, the experimental design is highly flawed because of the lack of a control group composed of cows not at risk of ketosis syndrome. This would better show the difference in breath acetone in healthy cows from ketotic cows (clinical or sub-clinical). The remaining part of the study about the relationships between blood BHB and urine/milk ketone bodies is already well-investigated in the previous literature with higher numbers of animals and samples, also the comparison between rapid test and lab test of blood BHB.

Specific comments are listed below.

L28-136: The introduction is well-written but too long. You should cut the redundancies and reduce the length.

L144-145: Why did you include only cows presenting a risk factor for ketosis? It should be present a control group of cows not at risk for comparison.

L143-157: You should include a descriptive table showing the number of cows presenting the specific criteria with mean and SD of the criteria measurement (e.g., n. of cows with BCS>3.75, mean and SD of BCS).

L223: Which is the reason to multiply the estimated read value by 0.4? Please explain.

L229: As for 0.4, why in this line is it elevated to (2-1)? Please explain.

L258-259: For validating the acetone sampling method, I would suggest providing also a coefficient of repeatability, which is a good indicator of the agreement between replicates.

L298-299: at L210 you said that two gas detection tubes were used per bag to measure acetone in breath. Here you are saying that you used a total of 3*3 test tubes. There is something wrong or the sampling is not well explained in the M&M. Please clarify.

L303: Please avoid single-sentence paragraphs. Here and in other parts of the manuscript.

L393-394: “All cows in the present study showed acetone concentrations >4 ppm”. Could it be due to the fact that only cows with high risk of ketosis were included in the study?

L415-417: Please check font size.

Table 1: I suggest substituting the table of maximum values with a more descriptive one with n. of samples, mean±SD, min and max values for the range. 

Figure 4: Read value acetone, units? Please add.

Figure 9b: missing units on acetoacetate y-axis. Please add.
